# Molecular Correlates of Venous Thromboembolism (VTE) in Ovarian Cancer

**DOI:** 10.3390/cancers14061496

**Published:** 2022-03-15

**Authors:** Deanna Glassman, Nicholas W. Bateman, Sanghoon Lee, Li Zhao, Jun Yao, Yukun Tan, Cristina Ivan, Kelly M. Rangel, Jianhua Zhang, Kelly A. Conrads, Brian L. Hood, Tamara Abulez, P. Andrew Futreal, Nicole D. Fleming, Vahid Afshar-Kharghan, George L. Maxwell, Thomas P. Conrads, Ken Chen, Anil K. Sood

**Affiliations:** 1Department of Gynecologic Oncology and Reproductive Medicine, The University of Texas M.D. Anderson Cancer Center, Houston, TX 77030, USA; dglassman@mdanderson.org (D.G.); kshea@mdanderson.org (K.M.R.); nfleming@mdanderson.org (N.D.F.); 2Cancer Biology Program, The University of Texas Graduate School of Biomedical Sciences, Houston, TX 77030, USA; afutreal@mdanderson.org (P.A.F.); vakharghan@mdanderson.org (V.A.-K.); kchen3@mdanderson.org (K.C.); 3Gynecologic Cancer Center of Excellence, Department of Gynecologic Surgery and Obstetrics, Uniformed Services University of the Health Sciences, Walter Reed National Military Medical Center, Bethesda, MD 20889, USA; batemann@whirc.org (N.W.B.); kconrads@whirc.org (K.A.C.); hoodb@whirc.org (B.L.H.); abulezt@whirc.org (T.A.); george.maxwell@inova.org (G.L.M.); conrads@whirc.org (T.P.C.); 4Henry M. Jackson Foundation for the Advancement of Military Medicine, Inc., Bethesda, MD 20817, USA; 5Department of Systems Biology, The University of Texas M.D. Anderson Cancer Center, Houston, TX 77030, USA; slee29@mdanderson.org; 6Department of Genomic Medicine, The University of Texas M.D. Anderson Cancer Center, Houston, TX 77030, USA; lzhao7@mdanderson.org; 7Department of Molecular and Cellular Oncology, The University of Texas M.D. Anderson Cancer Center, Houston, TX 77030, USA; jyao1@mdanderson.org; 8Department of Bioinformatics and Computational Biology, The University of Texas M.D. Anderson Cancer Center, Houston, TX 77030, USA; ytan1@mdanderson.org (Y.T.); jzhang22@mdanderson.org (J.Z.); 9Department of Experimental Therapeutics, The University of Texas M.D. Anderson Cancer Center, Houston, TX 77030, USA; crismonamirc@googlemail.com; 10Section of Benign Hematology, The University of Texas M.D. Anderson Cancer Center, Houston, TX 77030, USA; 11Women’s Health Integrated Research Center, Women’s Service Line, Inova Health System, Falls Church, VA 22042, USA

**Keywords:** venous thromboembolism, ovarian cancer, genomics, proteomics, genetic markers

## Abstract

**Simple Summary:**

Ovarian cancer is the most lethal form of gynecologic cancer. The prognosis is worse for patients with ovarian cancer who also develop a blood clot in their legs or lungs, known as a venous thromboembolism (VTE). Between 10-30% of women with ovarian cancer will develop a VTE but the mechanism behind this high incidence is not well understood. The aim of this study was to assess for the presence of tumor-specific molecular markers of VTE in order to better understand the relationship between ovarian cancer and VTE. We performed a comprehensive analysis assessing the proteomic and genomic profiles of 32 patients with ovarian cancer and identified that there are molecular differences in patients with a VTE compared to those without.

**Abstract:**

Background: The incidence of venous thromboembolism (VTE) in patients with ovarian cancer is higher than most solid tumors, ranging between 10–30%, and a diagnosis of VTE in this patient population is associated with worse oncologic outcomes. The tumor-specific molecular factors that may lead to the development of VTE are not well understood. Objectives: The aim of this study was to identify molecular features present in ovarian tumors of patients with VTE compared to those without. Methods: We performed a multiplatform omics analysis incorporating RNA and DNA sequencing, quantitative proteomics, as well as immune cell profiling of high-grade serous ovarian carcinoma (HGSC) samples from a cohort of 32 patients with or without VTE. Results: Pathway analyses revealed upregulation of both inflammatory and coagulation pathways in the VTE group. While DNA whole-exome sequencing failed to identify significant coding alterations between the groups, the results of an integrated proteomic and RNA sequencing analysis indicated that there is a relationship between VTE and the expression of platelet-derived growth factor subunit B (PDGFB) and extracellular proteins in tumor cells, namely collagens, that are correlated with the formation of thrombosis. Conclusions: In this comprehensive analysis of HGSC tumor tissues from patients with and without VTE, we identified markers unique to the VTE group that could contribute to development of thrombosis. Our findings provide additional insights into the molecular alterations underlying the development of VTE in ovarian cancer patients and invite further investigation into potential predictive biomarkers of VTE in ovarian cancer.

## 1. Introduction

The relationship between venous thromboembolism (VTE) and cancer was first reported by Trousseau in 1865, and VTE is now detected in at least 20% of patients with cancer [1,2,3,4,5]. A concurrent diagnosis of VTE portends a poorer prognosis [6], and apart from metastatic disease, VTE is the leading cause of death in cancer patients [7]. When comparing solid tumors, the incidence of VTE is among the highest in patients with ovarian cancer [8], with estimates ranging from 10–30% [9,10,11,12,13].

Several aspects of the phenotype of ovarian cancer and the way it is treated may explain, in part, the increased risk of VTE in this population. For example, patients often present with advanced disease, which manifests as a bulky tumor burden in the pelvis and a large volume of ascites that impedes venous drainage from the lower extremities, thereby increasing the risk of VTE [14]. Additionally, the backbone of treatment for advanced-stage ovarian cancer includes tumor reductive surgery requiring a prolonged exploratory laparotomy of the abdomen and pelvis, a known risk factor for thromboembolic events [14].

Beyond these risk factors that are unique to ovarian cancer, VTE rates are known to vary substantially based on ovarian tumor subtype and histology [11]. While ovarian cancer is made up of a heterogeneous group of histological subtypes, the majority originate from epithelial cells with two of the most aggressive types being clear cell (CC) and high-grade serous carcinoma (HGSC) [15]. The rates of VTE are highest in CC carcinoma followed by HGSC subtypes, suggesting the presence of a tumor-specific mechanism of thrombosis [16,17]. Repeated studies have demonstrated that the increased incidence of VTE in these populations is likely linked to the dynamic regulation of molecular factors involved in coagulation and fibrinolysis [18].

Currently, elevated levels of tissue factor and D-dimer in the serum are suggested to be related to a hypercoagulable state in patients with ovarian cancer [2,5,13,19,20]. Clinical risk calculators including the Khorana score, which accounts for patient factors such as BMI and cancer diagnosis, are utilized in combination with serum tests such as D-dimer to estimate the risk of VTE in patients with ovarian cancer [8]. However, the clinical utility of this score is limited, particularly in predicting VTE in the setting of an active malignancy such as ovarian cancer [21,22,23]. Previous investigations have evaluated potential genotypic relationships between cancer and concurrent VTE on a global scale by combining multiple cancer types, which all possess a variety of disease- and treatment-specific risk factors [2,3]. Moreover, focused analyses aimed at identifying specific genetic polymorphisms that might contribute to VTE have also been performed [24]. However, the call for the identification of specific molecular markers within tumor tissues that could advance both diagnostic and prognostic assessment in ovarian cancer has yet to be answered. We hypothesized that distinct molecular markers are present in HGSC tumors from patients who develop VTE compared to those who do not.

In this study, we carried out a detailed analysis of primary and metastatic tumors from patients with HGSC with or without VTE. We subjected the tissue samples from these clinically defined groups to multiplatform omics analysis, including whole-exome sequencing, RNA sequencing (RNA-seq), immune cell profiling, and a mass spectrometry (MS)-based proteomic assessment followed by fully integrated analyses.

## 2. Methods

In this retrospective case-control study, we searched the Ovarian Cancer Moon Shot database at the University of Texas MD Anderson Cancer Center (Houston, TX, USA) for patients that had a diagnosis of both HGSC and VTE who also had frozen tissue available for analysis. The database comprises 1314 patients with advanced ovarian cancer treated between 2013 and 2019 at our institution. Of these, only sixteen patients met the inclusion criteria to be able to be analyzed. To minimize selection bias for our control group, we searched the same database and identified 16 patients without a diagnosis of VTE. The control patients were matched on the basis of age (±10 years), BMI (±10 kg/m^2^), and tumor histology type. All patient information and tumor tissues were able to be analyzed; however, paired serum samples were available in only 24 cases (12 control and 12 VTE) and were used for matched germline testing. A control dataset composed of 420 samples extracted from XENA (https://xena.ucsc.edu, accessed on 1 January 2018), which included 88 patients from the Cancer Genome Atlas (TCGA) database, was utilized for the germline comparison for the eight unpaired samples (four control and four VTE). Clinical characteristics were abstracted from the electronic medical record for all patients by a single author (D.G.). The study was conducted under institutional review board approval #LAB10-0850.

### 2.1. RNA Sequencing

Frozen tumor samples were collected from the Institutional Tissue Bank, and RNA extraction was performed by the Biospecimen Extraction Resource of MD Anderson Cancer Center. In brief, total RNA was prepared using the Qiagen RNA Easy Mini Kit (QIAGEN, ThermoFisher Scientific, Inc., Valenica, CA, USA). The extracted RNA was then divided for complete RNA sequencing analysis and proteomic analysis. RNA sequencing was carried out by Novogene (Sacramento, CA, USA) on all samples. Bioinformatics analyses were subsequently performed to identify genes and pathways that may contribute to the development of VTE. A proteomic analysis was performed at the Women’s Health Integrated Research Center at Inova Health System. Statistical variance was determined using Fisher’s exact test with a maximum significant *p* value of 0.05.

### 2.2. Whole Exome Sequencing

Genomic DNA was prepared by the Biospecimen Extraction Resource of MD Anderson Cancer Center from 32 frozen tumor tissues from 32 patients and matched blood samples as germline controls. In brief, genomic DNA was extracted from frozen tissues and blood using a Qiagen DNA mini Kit and QIAamp DNA Blood Mini kit (QIAGEN, Valenica, CA, USA). DNA was diluted to 20 ng/µL in 55 µL and was transferred to microcentrifuge tubes. Whole exome sequencing was carried out by Novogene (Sacramento, CA, USA) on all samples. Tumor purity was assessed via Texomer [25]. Quality control of the raw data in FASTQ was performed by mapping to the human genome (hs37d5) using BWA info BAM files, which had been recalibrated for mark duplication and base quality score recalibration using Picard. Somatic mutations were called using varscan2 software on paired samples in VCF files. We applied the varscan2 software to detect somatic mutations based on the unpaired samples and the pooled control samples. A set of known germline variants from GATK was used as an additional filter by comparing the results of the somatic mutation burden from varscan2. All VCF files were annotated by VEP (ensemble.org, accessed on 21 December 2021). Mutation frequencies were compared via Fisher’s exact test by utilizing mafCompare software within maftools (bioconductor.org, accessed on 21 December 2021) to calculate significance. Copy number alterations were identified using DNAcopy (bioconductor.org, accessed on 21 December 2021). A cut-off of 0.5 was applied to identify copy losses, and a cut-off of two was used to identify copy gains.

### 2.3. Multiplexed, Quantitative Proteomic Data Analysis Using Tandem Mass Tags (TMT) and Data Processing Pipeline for Global Proteome Analyses

All 32 frozen tumor tissues from 32 patients diagnosed with high-grade serous ovarian cancer (HGSC) were prepared for and analyzed by a multiplexed, quantitative proteomics workflow as previously described by Lee, et al. [26]. Briefly, laser microdissection was used to isolate whole tumor collections (cancer and stromal cells combined), and samples underwent pressure-assisted digestion employing a barocycler (2320EXT Pressure BioSciences, Inc., South Easton, MA, USA) and a heat-stable form of trypsin (SMART Trypsin, ThermoFisher Scientific, Inc., Waltham, MA, USA). Peptide digest concentrations were determined using the bicinchoninic acid assay (BCA assay, Waltham, MA, USA), and ten micrograms of total peptide were labeled per tandem mass tag channel (TMTpro 16-plex, ThermoFisher Scientific, Inc., Waltham, MA, USA). Sample multiplexes were separated offline using basic reversed-phase liquid chromatographic fractionation on a 1260 Infinity II liquid chromatograph (Agilent, Santa Clara, CA, USA) into 96 fractions using a linear gradient of acetonitrile (0.69% min) followed by concatenation into 24 pooled fractions. Each pooled fraction was resuspended in 100 mM NH4HCO3 and analyzed by LC-MS/MS employing a nanoflow LC system (EASY-nLC 1200, Thermo Fisher Scientific, Inc., Waltham, MA, USA) coupled online with a Q-Exactive HF-X (Thermo Fisher Scientific Inc., Waltham, MA, USA). In brief, each fraction (~500 ng total peptide) was loaded on a nanoflow HPLC system fitted with a reversed-phase trap column (Acclaim PepMap100 C18, 20 mm, nanoViper, Thermo Scientific) and a heated (50 °C) reversed-phase analytical column (Acclaim PepMap RSLC C18, 2 µm, 100 Å, 75 µm × 500 mm, nanoViper, Thermo Fisher Scientific) coupled online with the MS. Peptides were eluted using a linear gradient of 2% mobile phase B (95% acetonitrile, 0.1% formic acid) to 32% mobile phase B over 120 min at a constant flow rate of 250 nL/min. High resolution (R = 60,000 at *m*/*z* 200) broadband (*m*/*z* 400–1600) mass spectra (MS) were acquired, followed by selection of the top 12 most intense molecular ions in each MS scan for high-energy collisional dissociation (HCD). Q Exactive HF-X–Full MS parameters were: AGC, 3 × 10^6^; RF lens, 40%; max IT, 45 ms; charge state, 2–4; dynamic exclusion, 10 ppm/20 s; MS2: AGC, 1 × 10^5^; max IT, 95 ms; resolution, 45 k; quadrupole isolation, 1.0 *m*/*z*; isolation offset, 0.2 *m*/*z*; NCE, 34; first mass, 100; intensity threshold, 2 × 10^5^; TMT optimization. Peptide and global protein-level identifications were generated by searching raw data files with a publicly available, non-redundant human proteome database (Swiss-Prot, Homo sapiens (http://www.uniprot.org, accessed on 18 December 2021) using Mascot (Matrix Science, Boston, MA, USA), Proteome Discoverer (Thermo Fisher Scientific Inc., Waltham, MA, USA) and in-house tools using identical parameters as previously described [26]. Differential analyses of global proteome or transcriptome matrices were performed using the LIMMA package (version 3.8) in R (version 3.5.2), and pathway analysis was performed using Ingenuity Pathway Analysis (Qiagen, Hilden, Germany) or Metascape (https://metascape.org/gp/index.html#/main/step1, accessed on 18 December 2021) using default parameters.

### 2.4. Immune Profiling Analysis

Immune profiling analysis was performed as previously described [26]. Immune contextures were estimated for the tumor samples by using the CIBERSORT tool (https://cibersort.stanford.edu, accessed on 23 October 2021). The algorithm was utilized to infer the fractions of the 22 immune cells relative to the total immune cell population. Comparisons were drawn between abundance and distribution of immune infiltrates for the VTE group (*n* = 16) and non-VTE control (*n* = 16) group.

## 3. Results

### 3.1. Patient Data

The 32 tumor tissue samples for this retrospective case-control study were analyzed in two groups: VTE (*n* = 16) and non-VTE controls (*n* = 16), as outlined in Figure 1. The demographic and clinical characteristics of these patients are described in Table 1. In the 16 VTE cases, the diagnosis of VTE occurred at various time points. Nine patients were diagnosed with a VTE simultaneously with their ovarian cancer diagnosis (56%), two patients were diagnosed with VTE during neoadjuvant chemotherapy (NACT, 13%), and five patients were diagnosed with VTE within 28 days of surgery or during adjuvant chemotherapy (31%). The median overall survival (OS) of patients with VTE was 2.17 years (SD = 1.35, *p* < 0.05). OS could not be calculated for the control group, as more than 80% of patients (*n* = 13, 81%) were still alive after a mean of 3.5 years (SD = 1.07). In this small cohort, the diagnosis of VTE represented a four-fold increased risk of death compared to patients without a VTE (log-rank hazard ratio = 4.37; *p* = 0.01).

### 3.2. Whole Exome Sequencing (WES)

This study included paired DNA samples representing germline samples and somatic tumor tissues from 32 patients, when available. In the VTE and non-VTE control groups, germline samples were unavailable for four and five patients, respectively. One tumor tissue sample from each group was of poor quality and was excluded after failing the quality control analysis as outlined in the methods. Therefore, the final analysis was performed with 11 paired samples in the VTE group and 10 paired samples in the non-VTE control group. The average somatic coverage was 50×. We examined copy number alterations across the exome and found no significant differences between the two groups. The most frequent copy number alterations in the VTE group were amplifications in *NBPF20* and deletions in *EBF2*, *FAM47A*, and *GTDC1*. In contrast, the most frequent copy number alterations in the non-VTE control group were deletions in *ANK1*, *ARHGAP39, FRMPD4*, and *MUC12*. There were no significant differences in the frequency of mutations observed between VTE and non-VTE samples.

### 3.3. RNA-Seq Identification of Differentially Expressed Genes

To investigate differences in gene expression between the groups, we performed an unsupervised hierarchical clustering on the basis of the 3000 most variable genes, and no dominant clusters were noted. Quality control assessment and principal component analysis showed no evidence of a batch effect based on clinical condition, tissue type, batch number, or tissue source. We identified 192 differentially expressed genes (DEGs), among which 130 were upregulated in the VTE group and 62 were downregulated. Specifically, the *F3* gene that encodes tissue factor (TF), a major factor involved in the pro-coagulant activity of cancer, was overexpressed approximately 4.5-fold in the VTE group vs. the control group (log2 fold change = 2.16, *p* < 0.0001; Figure 2A).

We also performed an outcomes analysis that utilized a dataset of ovarian cancer patients from The Cancer Genome Atlas (TCGA) to cross-compare the germline mutation rate of the DEGs in our discovery findings in ovarian cancer patients to patients without ovarian cancer. In doing so, we found that six of the identified DEGs were overexpressed in patients with ovarian cancer compared to those without and that these six DEGs were also consistently expressed in higher abundance in the VTE cohort compared to the non-VTE control cohort: *MT1M, CHIT1, EPHX3, RGS2, RASD1*, and *COL24A1*. A Cox regression analysis using age, grade, and gene expression indicated that 13 of the identified DEGs in the VTE group were associated with poorer overall survival. Specifically, *GAPDHP65* and *NRSN2-AS1* were found to have a hazard ratio of >1.5 (*p* = 0.018 and *p* < 0.001, respectively). Next, by gene set enrichment analysis, we found that the inflammatory cytokine TNF-α signaling pathway and the epithelial–mesenchymal transition pathway, which promotes tumor invasion and metastasis, were both upregulated in the VTE group (Figure 2B).

### 3.4. Differential and Integrated Analysis of Proteomic Data

We performed a proteomic analysis of the two sample groups utilizing a microscaled tandem mass tag (TMT) MS-based workflow and quantified over 7300 proteins (Appendix A). Differential analysis identified 255 significantly altered proteins in the VTE group compared to the control group (*p* < 0.01, Figure 3A, Appendix A). A principal component analysis showed that these altered proteins were responsible for 47.5% of the variance between groups (Appendix A). Additionally, after performing a supervised cluster analysis including VTE timing metadata, we found that these alterations did not demonstrate a direct relationship with the onset of VTE timing, suggesting that they may be intrinsically tumor-mediated and independent of ovarian cancer treatment status (Figure 3A).

Upon review of the significantly altered putative drug targets, we found that platelet-derived growth factor subunit B (PDGFB) was elevated nearly 10-fold in the VTE group compared to the control group (LIMMA, *p* < 0.001, Table 2). Pathway analyses of the altered proteins revealed that core extracellular matrix (ECM) canonical pathways were elevated, and carbon metabolism was reduced in the VTE group (Appendix A). The proteins that were elevated in the VTE group were biased toward expression within the extracellular space (52% vs. 0%, Figure 3B).

For cross-validation, we performed an integrated analysis with the RNA-seq data. This analysis compared co-quantified protein/transcript pairs between the VTE and control groups and identified a strong positive correlation (*n* = 67 protein/transcript pairs, Spearman Rho = 0.67, *p* < 0.0001; Figure 4A). We also observed a profound trend at the protein level in which collagen 1A1 and 3A1 were significantly co-altered in the VTE group compared to the control group (*p* < 0.001, Figure 4B,C).

### 3.5. Deconvolution Analysis of Immune Cell Profiles

To evaluate for differences in immune cell profiles between groups, we performed a cellular deconvolution assay of immune cell subsets between the VTE and control groups. Regulatory T cells (T_regs_) were found to be more abundant in the VTE group compared to the control group (4% vs. 1.3%, *p* = 0.014), while there were no significant differences in the proportion of CD4 or follicular helper T cells, respectively, between groups (18% vs. 15%, *p* = 0.35; 5.6% vs. 8.5%, *p* = 0.080). Resting, or non-activated, macrophages (M0) were higher in the VTE group compared to the control group (12.4% vs. 3.7%, *p* = 0.042). However, there were no differences in the differentiated pro-inflammatory (M1) or anti-inflammatory (M2) macrophages between groups (2.2% vs. 4.8%, *p* = 0.063; 2.2% vs. 2.3%, *p* = 0.606). There were also no statistically significant differences in the proportion of B cells or plasma cells between groups (11% vs. 9%, *p* = 0.39; 0.03% vs. 0.05%, *p* = 0.851).

## 4. Discussion

We report the results of a comprehensive multi-omics analysis of HGSC samples from both primary and metastatic tumor sites, delineated on the basis of a diagnosis of VTE. Our findings provide insight into potential tumor-mediated molecular factors that may predispose patients with HGSC to the development of VTE and serve as potential biomarkers for VTE. Given the lack of significant differences between the groups in whole-exome sequencing, our findings support the suggestion that while there are genetic differences in tumor tissues between these cohorts, the differences exist at the protein and transcript levels rather than upstream at the DNA level.

We observed that inflammatory and coagulation pathways were consistently elevated in the VTE samples compared to the control samples (Figure 2B). This suggests that there may be tumor-specific factors that lead to the development of VTE. The immune cell population differences between groups also suggest that there may be tumor-specific alterations in the inflammatory milieu in patients that develop a VTE compared to those with an absence of a VTE. We observed an increase in resting (M0) macrophages as well as T_regs_ in the VTE group compared to the control group. Monocyte-related inflammatory changes have been proposed as a mechanism of VTE development in multiple large-scale analyses [27,28], whereas the accumulation of T_regs_ in venous blood clots has recently been identified in mouse models as important in regulating thrombolysis via altering the recruitment and differentiation of monocytes [29].

We also observed significantly higher expression of tissue factor (TF), encoded by the gene *F3*, in the VTE samples compared to controls (*p* < 0.0001). While it is known that TF is a transmembrane protein expressed on monocytes in the vascular adventitia to allow for rapid activation of the coagulation cascade at the time of vascular injury, it is also known to be expressed and released by cancer cells [30]. Serum levels of TF are low in healthy individuals but can be increased in cancer as a result of cancer cells releasing microparticles containing TF [30]. TF has been implicated in the increased incidence of VTE observed in ovarian cancer and has been shown to be correlated among all histologic subtypes [13,18]. Our cohort was restricted to patients with high-grade serous histology, and we observed that TF was nearly 4.5-fold overexpressed in the VTE group compared to the control group (*p* < 0.0001, log2foldchange = 2.16). This suggests that TF may be of particular importance in serous histology, although a direct comparison with other histologies was not performed in this analysis.

At the protein level, platelet-derived growth factor subunit B (PDGFB) and its receptor (PDGFRB) were highly altered in the VTE cohort compared to controls. It is well known that PDGFB is expressed in the endothelium and on platelets and that it is a key element in thrombosis development [19]. Additionally, PDGFB has been implicated as a potential biomarker in the general population that is more specific than serum D-dimer assessment [19,20]. Our findings add to the body of evidence surrounding the importance and diagnostic potential of PDGFB in VTE.

Additionally, our findings highlight a relationship between VTE and the expression of both collagen 1A1 and 3A1 in tumor tissue. Collagen types I-III are extracellular proteins implicated in thrombosis as a result of their role in platelet adhesion and activation at the site of endothelial injury, where the ECM within the vascular wall is exposed to the bloodstream [31]. During tumor development, remodeling of the ECM is continually occurring and fragments of the ECM can be detected in the serum [32,33]. Collagen fragments can serve as “protein fingerprints” of an underlying ovarian malignancy [32]. In our investigation, we identified that collagen 1A1 and 3A1 are expressed predominately within the extracellular space of the tumor tissue of patients with VTE. Our findings support the concept that protein fragments could be shed during ECM remodeling of malignant tumors, which could then activate thrombogenesis even in the absence of a vascular injury or other provocation for VTE development. ECM fragments specifically related to type 1 collagen (C1M) have been associated with mortality and the development of metastatic disease and have been found to discriminate between the serum of patients with ovarian cancer and that of healthy controls [32,33,34,35]. This supports future investigation into ECM fragments as biomarkers for VTE in ovarian cancer.

The predominant strength of this investigation is the comprehensive nature of the multiplatform omics analysis with integration of the transcriptomic and proteomic data as well as a sub-population analysis of the immune cell distribution. Although the small sample size limits our findings, the distinct features identified between groups warrant examination on a larger scale.

## 5. Conclusions

We performed a comprehensive multiplatform omics analysis of HGSC tumor tissues from patients with and without VTE and identified makers unique to the VTE group that could contribute to development of thrombosis and the associated morbidity and mortality in ovarian cancer patients. Our findings provide additional mechanistic insight into the high prevalence of VTE in HGSC and invite further investigation into potential predictive biomarkers of VTE in ovarian cancer.

## Figures and Tables

**Figure 1 cancers-14-01496-f001:**
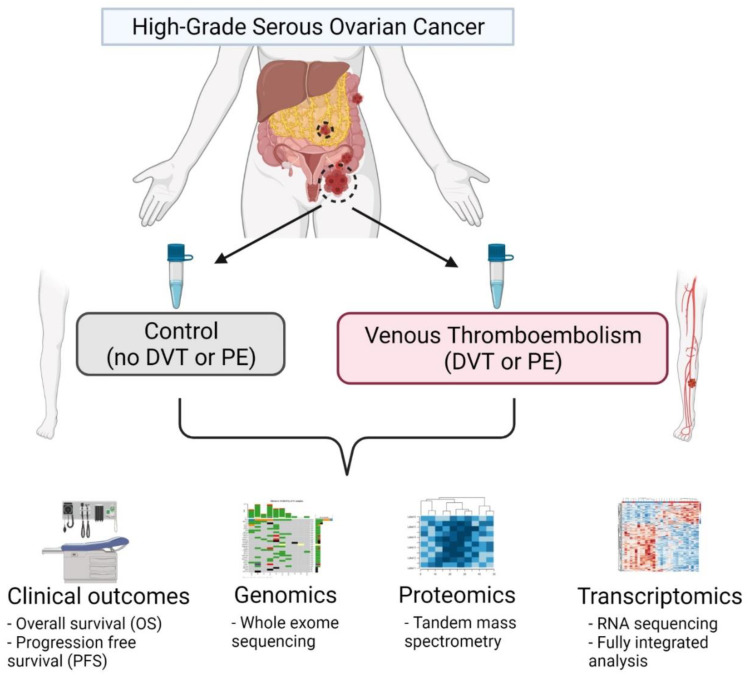
Schema of multi-omics analysis in HGSC patients with and without concurrent VTE. In this analysis, a total of 32 samples (*n* = 16 for both control and VTE groups) were obtained from the MD Anderson Cancer Center Ovarian Cancer Moon Shot database and subjected to a multi-platform omics analysis including whole exome sequencing (WES), quantitative mass spectrometry-based proteomics, and transcriptomics with RNA sequencing and a full integrated analysis incorporating clinical data and outcomes from the electronic medical record. Figure created with BioRender (https://app.biorender.com/illustrations/612d1ad2a3e2c500a5b6b6c3, accessed on 23 October 2021).

**Figure 2 cancers-14-01496-f002:**
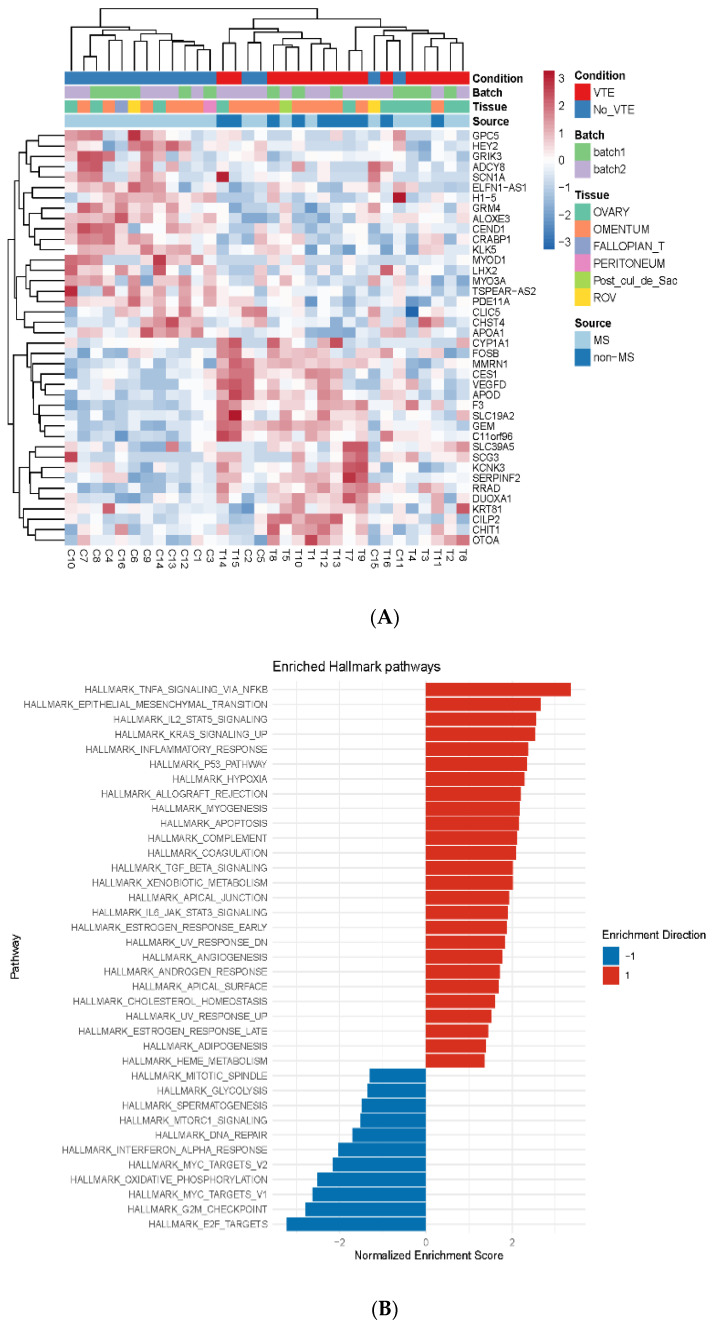
Differentially expressed genes and metabolic pathways in HGSC patients in VTE and control groups. (**A**) Heat map of the top 40 altered transcripts in HGSC tumor tissue in patients with and without venous thromboembolism (VTE). Abbreviations: Fallopian_T, fallopian tube; Post_cul_de_sac, posterior cul de sac; ROV, right ovary; MS, moonshot cancer database; non-MS, non-moonshot cancer database. (**B**) Gene set enrichment analysis (GSEA) identifying enriched pathways for VTE compared to non-VTE controls. Red indicates enhanced enrichment; blue indicates decreased enrichment.

**Figure 3 cancers-14-01496-f003:**
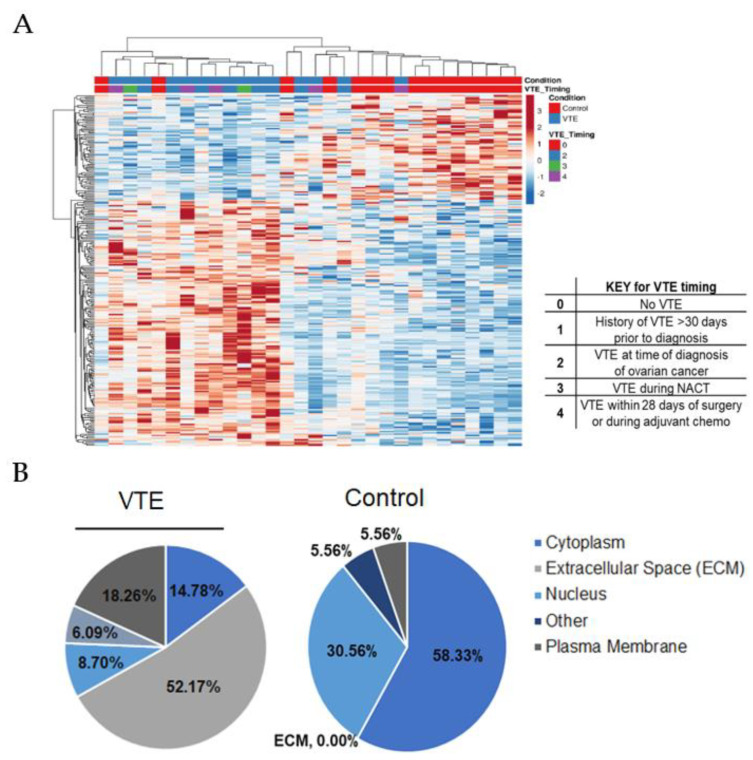
Significantly altered proteins in HGSC tumor tissues between VTE and control groups. (**A**) Supervised hierarchical cluster analysis of 255 significantly altered proteins between VTE and control tumors, including VTE timing metadata. (**B**) Cellular localization of 255 proteins that were significantly altered between VTE and control tumors (LIMMA, *p* < 0.01).

**Figure 4 cancers-14-01496-f004:**
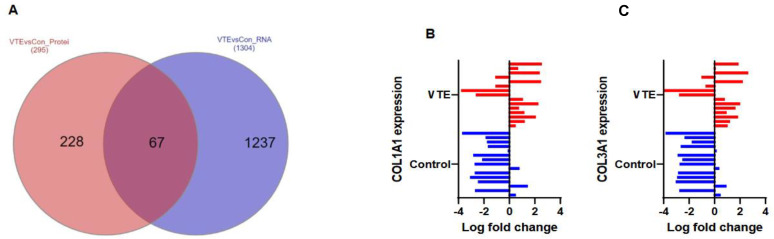
Co-altered proteins and transcripts between the VTE and control groups. (**A**) Comparison of co-quantified protein and transcript alterations in VTE vs. control tissue (LIMMA, *p* < 0.05, FC ± 1.5). (**B**) Differential protein expression matrix in collagen 1A1 between control (blue) and VTE (red); the *X*-axis represents log-fold change, *p* = 0.00093. (**C**) Differential protein expression matrix in collagen 3A1 between the control (blue) and VTE (red) groups, *p* = 0.00032.

**Table 1 cancers-14-01496-t001:** Patient demographics.

Variable	Control (Non-VTE)	VTE	*p*-Value
Age, mean (SD), years	59 (10.4)	66 (10.1)	0.09
BMI, mean (SD)	25.7 (6.6)	25.9 (6.3)	0.92
Baseline platelet count, mean (SD)	301 (101)	343 (151)	0.39
VTE timing, *n* (%)	N/A	16 (100)	-
At time of ovarian cancer diagnosis		9 (56)	-
During neoadjuvant chemotherapy		2 (13)	-
VTE within 28 days of surgery or during adjuvant therapy		5 (31)	-
Tissue source, *n* (%)			-
Primary tumor	7 (44)	7 (44)	1.0
Metastases	9 (56)	9 (56)	-
Overall survival	2.17	NYR *	-
Hazard ratio VTE vs. control		4.37	0.01

* NYR, not yet reached.

**Table 2 cancers-14-01496-t002:** Putative drug targets significantly altered in the VTE vs. control groups *.

Symbol	VTE vs. Control (Protein, log1.5FC)	Protein Name	Location	Drug(s)
PDGFB	3.089	platelet derived growth factor subunit B	Extracellular Space	sunitinib
COL3A1	2.428	collagen type III alpha 1 chain	Extracellular Space	collagenase clostridium histolyticum
COL1A1	2.236	collagen type I alpha 1 chain	Extracellular Space	collagenase clostridium histolyticum
COL1A2	2.007	collagen type I alpha 2 chain	Extracellular Space	collagenase clostridium histolyticum
CXCL12	1.805	C-X-C motif chemokine ligand 12	Extracellular Space	NOX-A12
APCS	1.772	amyloid P component, serum	Extracellular Space	dezamizumab
COL16A1	1.719	collagen type XVI alpha 1 chain	Extracellular Space	collagenase clostridium histolyticum
TNFSF13	1.703	TNF superfamily member 13	Extracellular Space	BION-1301
COL8A1	1.658	collagen type VIII alpha 1 chain	Extracellular Space	collagenase clostridium histolyticum
COL5A2	1.522	collagen type V alpha 2 chain	Extracellular Space	collagenase clostridium histolyticum
COL14A1	1.521	collagen type XIV alpha 1 chain	Extracellular Space	collagenase clostridium histolyticum
MMP28	1.481	matrix metallopeptidase 28	Extracellular Space	marimastat
TTR	1.233	transthyretin	Extracellular Space	tafamidis
APOC3	1.166	apolipoprotein C3	Extracellular Space	volanesorsen
COL6A1	1.154	collagen type VI alpha 1 chain	Extracellular Space	collagenase clostridium histolyticum
AOC3	1.152	amine oxidase copper containing 3	Plasma Membrane	ASP8232
DPP4	0.919	dipeptidyl peptidase 4	Plasma Membrane	saxagliptin
APOA1	0.903	apolipoprotein A1	Extracellular Space	ISIS 681257
PDGFRB	0.875	platelet derived growth factor receptor beta	Plasma Membrane	midostaurin
COL6A3	0.803	collagen type VI alpha 3 chain	Extracellular Space	collagenase clostridium histolyticum
BST1	0.803	bone marrow stromal cell antigen 1	Plasma Membrane	MEN1112
COL6A2	0.791	collagen type VI alpha 2 chain	Extracellular Space	collagenase clostridium histolyticum
ABCB1	0.609	ATP binding cassette subfamily B member 1	Plasma Membrane	dofequidar
PTGS1	−0.876	prostaglandin-endoperoxide synthase 1	Cytoplasm	sulindac/tamoxifen

* LIMMA *p* < 0.01, ±1.5-fold change.

## Data Availability

The data presented in this study are available on request from the corresponding author. The data are not publicly available to maintain confidentiality.

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
