# Peer review of "Molecular Correlates of Venous Thromboembolism (VTE) in Ovarian Cancer"

_cancers, 2022, doi:10.3390/cancers14061496_

Round 1

Reviewer 1 Report

This is a very interesting study evaluating differences in HGSOC patients with VTE or not based on a comprehensive mulit-omics analysis. The analysis provides some significant data for further evaluation, however it is significantly limited by the small sample size. In addition, the authors do not present basic clinicopathologic characteristics of the patients to reassure that they are equally distributed among groups. 

Author Response

Reviewer #1: This is a very interesting study evaluating differences in HGSOC patients with VTE or not based on a comprehensive mulit-omics analysis. The analysis provides some significant data for further evaluation, however it is significantly limited by the small sample size.

Response: We appreciate the thoughtful review and agree that the study findings are limited by the sample size and we have acknowledged this in the Discussion. However, given the comprehensive nature of this analysis with a multi-omics approach, we feel that the results are useful for the scientific community. 

In addition, the authors do not present basic clinicopathologic characteristics of the patients to reassure that they are equally distributed among groups. 

Response: Thank you for highlighting this. As shown in Table 1, the two groups were equally distributed in terms of age, BMI, and the tumor tissue source. We have also added the baseline platelet count data which, again, was not different between groups.

Reviewer 2 Report

The aim of this manuscript is to analyze the most significant molecular characteristics of primary and metastatic tumors, of patients with HGSC with or without venous thromboembolism (VTE).

Even if the manuscript provides an organic overview, with a densely organized structure and based on well-synthetized evidence, there are aspects to be mentioned, to make the article fully readable. For these reasons, the manuscript requires minor changes.

Please find below an enumerated list of comments on my review of the manuscript:

INTRODUCTION:

Some minor comments, for this introductive section. Specifically, Ovarian Cancer is the deadliest gynaecological malignancy. For this reason, the manscript will benefit from providing to the readers a brief and complete description of clinical and morphological characteristics of ovarian cancer, deeply linked to its molecular dynamics, as suggested by several and recent studies (see, for reference: Giusti, I., Bianchi, S., Nottola, S. A., Macchiarelli, G., Dolo, V. (2019). CLINICAL ELECTRON MICROSCOPY IN THE STUDY OF HUMAN OVARIAN TISSUES. EuroMediterranean Biomedical Journal14).

At the same time, Venous thromboembolism (VTE) is one of the major complications in cancer patients, often determinant for poor prognosis and short survival. In this context, genetic polymorphisms on specific genes, involved in the regulation of coagulation and fibrinolysis, are deeply associated to VTE, as reported by several functional and recent studies (see, for reference: Alsulaim AY, Azam F, Sebastian T, Mahdi Hassan F, AbdulAzeez S, Borgio JF, Alzahrani FM. The association between two genetic polymorphisms in ITGB3 and increase risk of venous thromboembolism in cancer patients in Eastern Province of Saudi Arabia. Saudi J Biol Sci. 2022 Jan;29(1):183-189. doi: 10.1016/j.sjbs.2021.08.073. Epub 2021 Aug 27. PMID: 35002407; PMCID: PMC8716864).

In conclusion, this manuscript is densely presented and well organized, based on well-synthetized evidences. The authors were lucid in their style of writing, making it easy to read and understand the message, portrayed in the manuscript. Besides, the methodology design was rigorous and appropriately implemented within the study. However, many of the topics are very concisely covered. This manuscript provided a comprehensive review of current knowledge in this field. Moreover, this research have futuristic importance and could be potential for future research. However, the minor concern of this manuscript is with the introductive section: for these reasons, I have minor comments only for the introductive section, for improvement before acceptance for publication. The article is accurate and provides relevant information on the topic and I suggest minor changes to be made in order to maximize its scientific impact. I would accept this manuscript, if the comments are addressed properly.

Author Response

Reviewer #2: The aim of this manuscript is to analyze the most significant molecular characteristics of primary and metastatic tumors, of patients with HGSC with or without venous thromboembolism (VTE).

Even if the manuscript provides an organic overview, with a densely organized structure and based on well-synthetized evidence, there are aspects to be mentioned, to make the article fully readable. For these reasons, the manuscript requires minor changes. Please find below an enumerated list of comments on my review of the manuscript: INTRODUCTION:

Some minor comments, for this introductive section. Specifically, Ovarian Cancer is the deadliest gynaecological malignancy. For this reason, the manscript will benefit from providing to the readers a brief and complete description of clinical and morphological characteristics of ovarian cancer, deeply linked to its molecular dynamics, as suggested by several and recent studies (see, for reference: Giusti, I., Bianchi, S., Nottola, S. A., Macchiarelli, G., Dolo, V. (2019). CLINICAL ELECTRON MICROSCOPY IN THE STUDY OF HUMAN OVARIAN TISSUES. EuroMediterranean Biomedical Journal14).

Response: Thank you for highlighting the importance of this. We have supplemented the introduction to include information on the histologic differences in ovarian cancer and the relationship between these subtypes and the development of VTE, focusing specifically on the link between the cancer subtypes and molecular pathways that regulate VTE.

At the same time, Venous thromboembolism (VTE) is one of the major complications in cancer patients, often determinant for poor prognosis and short survival. In this context, genetic polymorphisms on specific genes, involved in the regulation of coagulation and fibrinolysis, are deeply associated to VTE, as reported by several functional and recent studies (see, for reference: Alsulaim AY, Azam F, Sebastian T, Mahdi Hassan F, AbdulAzeez S, Borgio JF, Alzahrani FM. The association between two genetic polymorphisms in ITGB3 and increase risk of venous thromboembolism in cancer patients in Eastern Province of Saudi Arabia. Saudi J Biol Sci. 2022 Jan;29(1):183-189. doi: 10.1016/j.sjbs.2021.08.073. Epub 2021 Aug 27. PMID: 35002407; PMCID: PMC8716864).

Response: We appreciate your attention to this and have now added to the introduction to highlight the impact of genetic alterations and the development of VTE. We recognize that adding this information further emphasizes the importance of this study and the implications of identifying molecular markers unique to VTE.

In conclusion, this manuscript is densely presented and well organized, based on well-synthetized evidences. The authors were lucid in their style of writing, making it easy to read and understand the message, portrayed in the manuscript. Besides, the methodology design was rigorous and appropriately implemented within the study. However, many of the topics are very concisely covered. This manuscript provided a comprehensive review of current knowledge in this field. Moreover, this research have futuristic importance and could be potential for future research. However, the minor concern of this manuscript is with the introductive section: for these reasons, I have minor comments only for the introductive section, for improvement before acceptance for publication. The article is accurate and provides relevant information on the topic and I suggest minor changes to be made in order to maximize its scientific impact. I would accept this manuscript, if the comments are addressed properly.

Response: Thank you for your comments; we appreciate the positive feedback.
